# Laser Ablated Periodic Nanostructures on Titanium and Steel Implants Influence Adhesion and Osteogenic Differentiation of Mesenchymal Stem Cells

**DOI:** 10.3390/ma13163526

**Published:** 2020-08-10

**Authors:** Kai Oliver Böker, Frederick Kleinwort, Jan-Hendrick Klein-Wiele, Peter Simon, Katharina Jäckle, Shahed Taheri, Wolfgang Lehmann, Arndt F. Schilling

**Affiliations:** 1Department for Trauma Surgery, Orthopaedics and Plastic Surgery, University Medical Center Goettingen, Robert Koch Straße 40, 37075 Göttingen, Germany; katharina.jaeckle@med.uni-goettingen.de (K.J.); shahed.taheri@med.uni-goettingen.de (S.T.); Wolfgang.Lehmann@med.uni-goettingen.de (W.L.); arndt.schilling@med.uni-goettingen.de (A.F.S.); 2Laser-Laboratorium Göttingen e.V. (LLG), Hans-Adolf-Krebs-Weg 1, 37077 Göttingen, Germany; frederick.kleinwort@llg-ev.de (F.K.); jhkw@llg-ev.de (J.-H.K.-W.); peter.simon@llg-ev.de (P.S.)

**Keywords:** titanium, steel, implant, laser beam interference ablation, surface nano-topography, nano-topology, stem cell, bone, osteoblast, periodic nano-structures

## Abstract

Metal implants used in trauma surgeries are sometimes difficult to remove after the completion of the healing process due to the strong integration with the bone tissue. Periodic surface micro- and nanostructures can directly influence cell adhesion and differentiation on metallic implant materials. However, the fabrication of such structures with classical lithographic methods is too slow and cost-intensive to be of practical relevance. Therefore, we used laser beam interference ablation structuring to systematically generate periodic nanostructures on titanium and steel plates. The newly developed laser process uses a special grating interferometer in combination with an industrial laser scanner and ultrashort pulse laser source, allowing for fast, precise, and cost-effective modification of metal surfaces in a single step process. A total of 30 different periodic topologies reaching from linear over crossed to complex crossed nanostructures with varying depths were generated on steel and titanium plates and tested in bone cell culture. Reduced cell adhesion was found for four different structure types, while cell morphology was influenced by two different structures. Furthermore, we observed impaired osteogenic differentiation for three structures, indicating reduced bone formation around the implant. This efficient way of surface structuring in combination with new insights about its influence on bone cells could lead to newly designed implant surfaces for trauma surgeries with reduced adhesion, resulting in faster removal times, reduced operation times, and reduced complication rates.

## 1. Introduction

The World Health Organization reported distal radius fractures among the ten most expensive medical incidents worldwide. These fractures are most prevalent in women older than 50 years who suffer from osteoporosis [1,2,3]. An important early step for fracture healing is the immobilization of the broken bones, which provides support for the biological aspect of the bone healing process. There are several options to adjust and immobilize broken bones in their appropriate position such as screws, plates, Kirschner wires, cerclage wires, locking pins, or specialized devices such as external/internal fixators and intramedullary nails. For long bones, like the distal radius, intramedullary nailing is the gold standard for fracture management [4,5,6].

### 1.1. Steel and Titanium Implants

Currently, implants for such intramedullary nails are made of titanium or stainless steel [4]. Steel implants are still mainly used worldwide today, especially in the USA, due to their lower material cost [7]. However, clinical investigations point to a higher rate of allergy with steel as compared to titanium implants. Experimental studies also showed that bones treated with titanium are less susceptible to infection and have a better clinical outcome when infection occurs [7]. Furthermore, titanium has a lower susceptibility to corrosion and better biocompatibility. Due to these advantages, titanium is gradually replacing steel as osteosynthesis material, especially since improved manufacturing technology has decreased its cost. In Germany, titanium is already the main osteosynthesis material [8].

### 1.2. Implant Removal

After healing of the fracture, surgical removal of the immobilization aids is a common procedure; e.g., in 2010, about 180,000 metallic implants were surgically removed from the musculoskeletal system of German patients, excluding the removals conducted by general practitioners [9]. The removal of implants is a critical procedure [10]. This practice bears different risks to be considered, which can lead to complication rates up to 26% for the removal of the implanted material including: 6.8% subsequent wound infections, 11.5% peripheral nerve lesions, and a refracture rate of about 7.7% [10]. In addition, integration of the implanted material and the surrounding soft tissues makes implant removal somewhat difficult [11].

Steel implants are covered by a thin fibrous surface inside the body and are therefore a bit easier to remove than titanium implants, which integrate with direct contact to the bone [11]. Consistently, the average time for steel implant removal is 84 min as compared to 110 min for titanium implants [11]. In Germany, the cost of an operating hour is currently in the range of 300 € for personnel only, plus the costs for the necessary infrastructure.

As the tendency to stabilize bone fractures with surgical intervention is constantly increasing, the need for implant removal is increasing accordingly, which inflates the costs of bone fracture treatments [9]. Given the average cost of approximately 800 € for out-patient care and approximately 2000 € for short-term stationary interventions, the total costs of bone fracture treatment in Germany amounts to 250 million Euro per year [12] and an increasing number of bone fracture incidents are expected for the future. In parallel, the need for surgical implant removal in the western world has also been increasing [12] and it will almost certainly continue to do so in the future.

A possible approach to reduce overall costs could be reducing removal times by preventing material incorporation.

### 1.3. Laser Ablation Structuring

Laser surface texturing of metals is a precise and reliable method to modify properties of the surface to obtain new functional features without adding other materials or compounds [13,14]. Laser-based surface modifications have been studied intensively during the last decade [15]. Pico- and femtosecond laser technologies have opened new pathways towards modifying the surface structure with nanometric precision [16,17]. An interferometric processing technique enables the generation of structures well below the spot size of the laser. Furthermore, it allows for precise and fast fabrication of large surface areas [18], making it viable for industrial applications.

Here, we aimed to study whether the adhesion, morphology, and the osteogenic differentiation capacity of cells can be manipulated by nano-structuring the surface of implant materials (steel and titanium) via such laser treatments.

## 2. Materials and Methods

### 2.1. Surface Structuring of Implant Materials

The structuring of the sample surfaces was done by laser beam interference ablation, applying a special interferometer-based setup developed at Laser-Laboratorium Göttingen sketched in Figure 1. Third harmonic laser pulses from an ultrashort pulse laser (Hyper Rapid 50, Coherent, Dieburg, Germany) with a duration of ~8 ps at a wavelength of 355 nm and a repetition rate of 400 kHz were used for surface structuring. The pulses were guided through a diffractive laser scanner consisting of a fast galvanometric scanning head (intelliSCAN_de_ 14, SCANLAB GmbH, Puchheim, Germany), an objective lens, and a rotatable grating interferometer and focused onto the sample surface with a focus diameter of 30 µm. This configuration projected a well-defined, high-quality linear grating interference pattern with a period of 1.5 µm onto the workpiece where the shape of the structured area could be freely chosen by the scanning system while the direction of the grating lines could be precisely changed by rotating the interferometer around the optical axis. This allowed full control over the orientation of the grating lines without having to move the sample. Repeated irradiation of the same area at different grating orientations results in complex periodic patterns [19,20].

The depth of the resulting structures on the sample surface can be adjusted within a wide range through the energy of the laser pulses and the beam feed during the scanning process. This was used to realize five different structure depths (D1–D5) for each of the six periodic patterns (S1–S6), which in total resulted in 30 unique combinations. Each combination was denoted by two numbers: The first signifies the 6 different types of structures and the second number denotes the sequentially increasing depths of the grooves (e.g., 3–5 indicates the combination with the structure type number 3, and the deepest groove. Structures 1-D had a simple linear texture. Structures 2-D, 3-D, 4-D were one-fold crossed at respective angles of 45°, 67.5°, and 90°. Structure 5-D was twofold crossed with crossing angles of 60° and 120°. Structure 6-D was threefold crossed at 45°, 90°, and 135°. The structure types are illustrated in Figure 2.

For the initial testing, the stainless-steel samples (14 mm × 14 mm) were provided with 0.5 × 0.5 mm square test fields. For each of the 30 structure/depth combinations, 5 square fields were produced on each sample, resulting in 150 squares per sample (Figure 3, Appendix A). Each of the 5 identical fields was encircled by a fine line for easier evaluation of the samples. After initial testing for cell adhesion, structures 1-5, 2-5, and 5-5 were selected, and 14 mm × 14 mm stainless-steel as well as polished titanium samples were structured on the whole surface area (14 mm × 14 mm). An example of untreated and complete structured plates is depicted in Appendix A, C. In the first step, all plates were ground by sandpaper in increasing grids (400–1200). After each sandpaper alteration, the direction of grinding was changed. After grinding, the plates were polished with a felt disc and then with a cotton buff including diamond-paste with a constant change of direction. Since titanium is oxidized directly, it is necessary to polish it immediately after the last sanding process. Finally, polished plates were cleaned in an ultrasonic bath, dried under compressed air, and structured using laser beam ablation.

### 2.2. Analysis of the Surface Topology

The resulting surface structures on the stainless steel and titanium samples were analyzed using reflected-light microscopy, atomic force microscopy (Park Systems AFM XE-150, non-contact cantilever AR5-NCHR with high aspect ratio (>5:1) tip), and scanning electron microscopy (EVO MA10, Carl Zeiss Microscopy GmbH, Jena, Germany). The resulting AFM data were processed and analyzed using the Park Systems XEI 4.3 software.

### 2.3. Cell Culture

Single-cell-derived human mesenchymal stem cell lines expressing hTERT (SCP1 cell line [21], origin hMSCs (unmodified) were purchased from Cambrex Corporation (East Rutherford, NJ, USA)) were maintained in DMEM low glucose medium with 10% fetal calf serum (FCS) and 1% antibiotics (penicillin and streptomycin) according to standard conditions in humidified incubators at 37 °C and 5% CO_2_.

### 2.4. Lentivirus Production

Vesicular stomatitis virus glycoprotein (VSV-G) pseudotyped lentiviruses were generated in HEK293FT cells. Cells were seeded in 6-well plates at 1 × 10^6^ cells/mL and directly transfected with pLenti CMV GFP Neo [22] for GFP delivery, psPAX2 for viral capsid proteins, and pCMV-VSV-G [22] for viral envelope proteins (summarized plasmids are listed in Table 1). Also, 16 h after transfection, sodium butyrate containing medium (0.01 M) was added. After 8 h, medium without sodium butyrate was used and collected every 24 h. After 5 days, medium was centrifuged for 30 min at 2000× g, filtered through 450 nm filters (Sartorius, Germany), and concentrated via Vivaspin columns (Sartorius, Göttingen, Germany).

### 2.5. Generation of Stable SCP1-GFP Cell Lines

SCP1 cells were seeded in 24-well plates (50,000 cells/well). Then, 24 h later, the cells were infected with lentivirus carrying VSVG as an envelope protein and GFP as a gene of interest. Also, 96 h after infection, geneticin (G418, ThermoFisher Scientific, MA, USA) -containing medium was added and cells were selected for 14 days.

### 2.6. Cell Adhesion Determination

SCP1-GFP cells (140,000/plate) were seeded on structured steel plates (S1–S6, D1–D5, Figure 3A). 24 h after seeding, cell adhesion was quantified by manual counting of fluorescence cells (Figure 3B). Due to the varying cell density on the steel plate, the non-treated surface directly beside the structured surface was used for comparison. Experiments were performed in biological triplicates. Cell morphology was analysed via fluorescence microscopy using the Leica DMi8 platform (Leica, Germany).

### 2.7. Co-Culture Implants with Mesenchymal Stem Cells (MSCs) and Osteogenic Differentiation

SCP1 cells were cultured on the polystyrene surface (cell culture flasks), non-structured stainless steel, and titanium plates (14 mm × 14 mm). Furthermore, structured stainless steel and titanium plates (structures 1-5, 2-5, and 5-5) were used for comparison. Osteogenic differentiation was performed for 4 weeks with the osteogenic differentiation medium (Table 2). Co-culture experiments were performed in biological triplicates.

### 2.8. RNA Isolation, Reverse Transcription and Gene Expression Analysis

Total RNA was isolated via phenol/chloroform extraction according to the manufacturer’s protocol (Trizol, ThermoFisher Scientific, Waltham, MA, USA). RNA was measured by the DeNovix DS-11 FX+ System (DeNovix, Wilmington, NC, USA) and was then stored for further processing at −80 °C.

Reverse transcription was accomplished with 200 ng of total RNA using M-MLV Reverse Transcription Kit (Promega, Madison, WI, USA). QRT-PCR was performed on CFX96 Real-time PCR Detection System (Biorad, CA, USA) using SYBR Green (Biorad, Hercules, CA, USA) as a detection marker. Relative expressions of *osteocalcin* (*OCN*), *RUNX Family Transcription Factor 2* (*Runx2*), *Collagen 1*, *Collagen 2*, *Aggrecan*, *SRY-Box Transcription Factor 9* (*Sox9*) and *β-2-microglobulin* were measured in triplicates and calculated via the ΔΔCt-method [24] using *β-2-microglobulin* as a housekeeping gene. Primer sequences are depicted in Table 3.

### 2.9. Statistical Analysis

One-way analysis of variance (ANOVA) with the Tukey multiple comparison test was conducted for gene expression analysis, while the Kolmogorov-Smirnov normality test followed by an unpaired T-test was performed for cell adhesion comparison. Throughout the manuscript, data are given as mean ± standard deviation (SD), and *p* values < 0.05 were considered statistically significant. Statistical evaluation was carried out using GraphPad Prism 5.01 (GraphPad Software, CA, USA). Statistical significance was defined as follows: no significance (ns), *p* < 0.05 (*), *p* < 0.01 (**) and *p* < 0.001 (***).

### 2.10. Circularity Calculation

The cell morphology was quantified by calculating the circularity index (Circ.), which is defined by the following formula:$$\mathrm{Circ}.=4\pi \left(\frac{\mathrm{area}}{{\mathrm{perimeter}}^{2}}\right)$$

A Circ. of 1.0 indicates a perfect circle, while Circ. closer to 0.0 indicates an increasingly elongated polygon. The measurement was conducted for the untreated titanium and steel plates, as well as the structures 1-5, 2-5, and 5-5. Fluorescent images of the SCP1 cell lines 48 h after cell seeding were normalized with respect to color exposure and threshold, and were evaluated using the ImageJ software (Version 1.52a, NIH, USA) using the analyze particles function. For each structure type, at least 50 cells were measured, and mean values and standard deviations were calculated.

## 3. Results

### 3.1. Influence of Different Nano-Topographies on Cell Adhesion

An overview of a steel plate with 6 different structures and 5 altered depths with 5 replicas each is depicted in Figure 3A. Strong effects on cell adhesion were observed for deep structures (red square), while only minor effects were detected for shallow structures (blue square, Figure 3B).

Cell adhesions on untreated and structured plates were compared and analyzed. Overall, the strongest effects were observed for the deepest combinations (data not shown). Thus, we focused on these modifications (Figure 4). Significantly reduced cell numbers per mm^2^ were detected for the structures 1-5 (untreated 35 ± 2 cells/mm^2^, structured 22.3 ± 1.5 cells/mm^2^), 2-5 (untreated 86.6 ± 10.4 cells/mm^2^ compared to 59.7 ± 4.8 cells/mm^2^), 5-5 (70.3 ± 9 cells/mm^2^ compared to 47.7 ± 6.5 cells/mm^2^) and 6-5 (71.3 ± 4 cells/mm^2^ compared to 59.9 ± 1.5 cells/mm^2^). For the structures 3-5 (51 ± 5.6 mm^2^ compared to 49.3 ± 3.8 mm^2^) and 4-5 (62.7 ± 6.8 mm^2^ compared to 54.7 ± 4.5 mm^2^), no significant differences were measured.

After the initial tests for cell adhesion, the three structures with highest cell adhesion reduction 1-5, 2-5, and 5-5 were selected, and 14 mm × 14 mm polished stainless-steel as well as titanium samples were structured on the complete surface area. The samples were analyzed using scanning electron microscopy (Figure 5) and atomic force microscopy (Figure 6).

The scanning electron microscopy (SEM) pictures showed a highly consistent nano-topology of the periodic nanostructures on steel and titanium samples (Figure 5).

The atomic force microscopy (AFM) measurements revealed a range for the structural depths from approximately 200 nm (structure 1-1) to approximately 1400 nm (structure 5-5). Linear structures were measured in all structure depths (D1–D5) on steel and titanium plates and are depicted in Appendix A. Figure 6 shows the AFM measurements of the deeper ridges for the stainless-steel samples in types 1-5, 2-5, and 5-5. Adjustments for the laser parameters when switching from steel to titanium were made based on the line structures of medium depth 1-3. This apparently does not translate to the laser parameters needed for the deepest structures. The structure 1-5 had a line depth of approximately 880 nm on steel and approximately 700 nm on titanium. The crossed structures 2-5 and 5-5 had different depths along different symmetry axes due to multiple laser radiation in the intersection areas of the super-imposed line structures. For structure 2-5, the depths were approximately 750 and 990 nm on steel and 400 and 600 nm on titanium. For the structure 5-5, the depths were approximately 1050, 1300, and 1400 nm on steel and 900, 1200, and 1300 nm on titanium. These structure depths are summarized in Table 4.

### 3.2. Influence of Surface Structure on Cell Phenotype

The influence of surface structuring on cell phenotype is visualized in Figure 7. Non-treated, polished titanium and stainless-steel samples did not influence cell morphology of the SCP1 cells compared to polystyrene surfaces (data not shown). The structure 1-5 changed cell morphology to longer and thinner cell phenotypes, mimicking the linear, parallel structures on titanium and stainless-steel plates. The structure 2-5 showed only minor effects on cell morphology, while the structure 5-5 changed the cellular phenotype to smaller and rounded cells (Appendix A). For structures 3-5 and 4-5, no effects on cell phenotype were observed (data not shown).

### 3.3. Surface Modifications Lead to Osteogenic Inhibition of Human Mesenchymal Stem Cells (MSCs)

Next, we focused on the influence of surface modification on osteogenic differentiation of mesenchymal stem cells (MSCs, Figure 8). SCP1 cells were cultured on a polystyrene surface as a “standardized” control. After osteogenic differentiation, cells showed a significant increase of *Osteocalcin* (2.65 ± 0.86 fold), *Collagen 1* (1.74 ± 0.82 fold), and *RUNX2* (2.59 ± 1.42 fold) expression. Osteogenic differentiation on untreated stainless-steel plates displayed similar results; i.e., *Osteocalcin* (2.45 ± 1.45), *Collagen 1* (1.48 ± 0.48), and *RUNX2* (2.99 ± 1.28) expressions were significantly increased compared to the non-differentiated control group. For osteogenic differentiation on untreated titanium plates, a significant increase of *OCN* (2.45 ± 2.11) and *RUNX2* (2.59 ± 1.68) was detected, while *Collagen 1* (1.28 ± 0.73) was unaffected. We observed no influence of the material (polystyrene vs. stainless-steel vs. titanium) on *OCN*, *Collagen 1,* and *RUNX2* expression in non-differentiated control samples.

However, we observed a significant reduction of *OCN* expression on structured stainless-steel and titanium surfaces compared to non-structured surfaces for all tested structures except for structure 5-5 on titanium plates (Figure 8A). The reduction of *Collagen 1* expression was found for structures 2-5 and 5-5 on steel, while no significant difference was observed for structure 1-5. On surface-treated titanium plates, *Collagen 1* expression was not changed (Figure 8B). On the other hand, a strong reduction of *RUNX2* expression compared to non-structured controls was detected for all structure types on steel plates. On titanium plates, a significant reduction of *RUNX2* was observed for structure 1-5, while structure 2-5 and 5-5 did not influence *RUNX2* expression (Figure 8C).

To test if the decrease of osteogenic differentiation markers in combination with a round cell morphology could be a reflection of chondrogenic differentiation, we additionally examined the expression of chondrogenic-relevant genes (*Aggrecan*, *Sox9* and *Collagen 2*), but we did not find an indication that this was the case (Appendix A).

## 4. Discussion

In this study, the cellular response of human MSCs to periodic nano-topographies introduced using laser ablation has been analyzed on clinically-relevant osteosynthetic materials (steel and titanium surfaces).

Cell adhesion of human MSCs was significantly reduced by several configurations of surface modification (Figure 4). Three surface topographies that exhibited the highest reduction of cell adhesion were selected for further analysis (structures 1-5, 2-5, and 5-5). It was confirmed that cell morphology was altered markedly depending on the surface structure (Figure 7). This is in line with other reports where reduced cell adhesion and increased separation and orientation of murine fibroblasts were described on titanium-alloy substrates [25]. Cell adhesion is generally affected by surface proteins called cell-adhesion molecules (CAMs), which can be further categorized into integrins, immunoglobulin superfamily, cadherins, and selectins [26]. Each of these proteins recognizes different ligands, resulting in cell junctions [27]. Cells adhere to metal surfaces via focal contacts and extracellular matrix adsorption to the metal surface [25,28]. Structured metal surfaces can address three different aspects of cell adhesion [25]. First, mammalian cells generally exhibit limited flexibility, preventing the adaptation to sharp surfaces such as laser-induced spikes. Second, the CAMs are clustered to focal adhesions [29]. These focal adhesions have a sub-µm lateral dimension and typical distances up to a few µm [25]. Therefore, the contact area is limited to the spikes, which reduces the contact surface and therefore the possible adhesion force. Finally, surface treatment can lead to oxidation, limiting the adhesion of CAMs, thus reducing the cell adhesion [30].

Positive osteogenic differentiation upon upregulation of osteogenic markers was observed on polystyrene as well as on untreated titanium and steel surfaces (Figure 8A–C). Aside from reduced cell adhesion and morphological changes, decreased osteogenic differentiation capacities of human MSCs on several structured types were observed when compared to polystyrene and untreated metal surfaces. Modification of steel plates led to a reduction of *OCN* and *RUNX2* for all tested structure types compared to polystyrene, while *Collagen 1* expression was not affected by linear structures (i.e., structure 1-5).

Cells on linear structures showed a prolonged phenotype, which resembles fibroblasts. These cells are known to produce high levels of *Collagen 1* [31,32], which could explain the lack of a corresponding effect on *Collagen 1* expression.

Modification of titanium plates, on the other hand, resulted in marginal effects on osteogenic differentiation inhibition. In general, modification of steel plates generated stronger effects, which can be explained by deeper manufactured structures on steel plates. We examined the nano-topography of the implant materials after surface structuring using SEM and AFM techniques. The laser parameters were chosen to yield the same surface topography on both materials. While the SEM examinations showed only slight differences between the lateral appearance of the structures on the two surface-treated materials (Figure 5), AFM measurements revealed that the structures on steel plates were at least 100 nm deeper than that of titanium (Figure 6, Table 4). This can be explained by a higher ablation threshold of titanium compared to steel, which was not fully compensated by the chosen laser parameters.

Moreover, the osteogenic differentiation of human MSCs was inhibited in nearly all tested structures of steel, while only selected structures had an effect on titanium. This might be partially explained by the difference observed in the depth of the two materials. We also found a connection between structure depth, cell morphology, and adhesion. Further examination of even deeper structures is required in order to fully exclude an influence of the material like chemical potential, wettability, or a possible difference in oxidization due to the laser process.

Other studies on metal surface modifications and osteoblasts so far have concentrated on methods to increase osteogenic differentiation. Nano-modified titanium that is created by high temperatures is shown to exhibit osteoblast sensitivity in response to surface modifications [33,34]. It is also shown that surface modification through acid etching and electrochemical machining can induce osteoblast (MG63) sensitivity to submicron-scale architectures. MG63 cell morphology on anodized surfaces was reportedly similar to standard surfaces, while MG63 on etched surfaces had a more elongated shape [35]. A higher synthesis of alkaline phosphatase and deposition of calcium was also found after 21 and 28 days of cell culture in the presence of osteoblasts on nanophase ceramics [36]. While most of these studies have used immortalized osteoblast cell lines (i.e., osteosarcoma MG63), we differentiated the human mesenchymal stem cell line (SCP1 [21]) for 4 weeks into osteoblasts. This model is more time-consuming but possibly more suitable and relevant for mimicking osteoblast behavior. One piece of evidence supporting this claim is that former studies have found that MG63 cells show reduced ALP activity and a low mineralization rate [37,38], which indicates lower osteoblast-like properties.

Several studies have shown that nano-topographical patterns such as nano-pillar or nano-grill structures affect human MSC differentiation [39]. Compared to non-treated surfaces, nano-pillar and nano-hole topographies led to enhanced chondrogenic differentiation, while nano-grill topographies inhibited this differentiation route. Our study demonstrated the sensitive reaction of MSCs to surface composition and structure. Most surface treatments so far have mainly focused on the increase of surface area and roughness via random micro- and sub-microscale modifications [33,40]. Our approach, on the other hand, introduced repetitive periodic structures, leading to modified phenotypes and reduced cell adhesion.

Primary chondrocytes especially reveal a cobblestone-like phenotype. In particular, for structure 5-5, cobblestone-like phenotypes were observed for MSCs. Hence, we decided to check for chondrogenic gene expression as well (Appendix A). Our results demonstrated only a minor influence on chondrogenic differentiation. Mainly, *Aggrecan* was changed upon treatment, with the effect seemingly dependent on surface stiffness (i.e., metal vs. polystyrene) rather than surface modification. It is known that substrate stiffness is a potential factor for chondrogenic differentiation, with softer substrate stiffness preferred for chondrogenic differentiation [41]. Nevertheless, other factors such as surface charge, surface binding to oxygen, or chemical effects can influence cell behavior and differentiation. Further future studies could shed some light onto these observations.

Osseointegration and cell adhesion are critical processes for medical implantation [33]. Especially for metal implants in load-bearing regions, osseointegration-induced stabilization and accelerated tissue integration are crucial. Hence, several surface modification strategies that improve osseointegration have been performed in the last decade [42,43,44,45]. Especially for metal implants in load-bearing regions, increased bone formation and accelerated tissue integration seem reasonable. Nevertheless, several studies describe complications during nail removal through nail failure, jamming in the fracture callus during removal, or difficulties and need for special instruments [46,47,48]. Most studies of modified surfaces focused on increased cell adhesion/osseointegration [39,42,43,44,45] or decreased bacteria adhesion/prevention of biofilm formation [49,50,51]. Nevertheless, Lavenus et al. described limited effects of nanopore-modified titanium on the osteogenic differentiation of MSCs. Ti300 (titanium surface with 300 nm pores) showed reduced *OCN* and *Collagen 1* expression after 12 days of cultivation, while smaller nanopores (30–150 nm) showed opposite effects [52]. These observations support our results of prevented osteogenesis by deep surface structures (700–1400 nm).

Further research could focus on modifications of curved surfaces (e.g., intramedullary nails), as a surface modification of curved and non-plane structures is methodically more challenging but extremely clinically relevant. Removal of intramedullary nails can be challenging [53,54], and the process of removal can be facilitated by laser-ablated periodic structures, which would be an important step towards the future implementation of modified implants for daily use in orthopedics departments. Locally defined inhibited cell adhesion and therefore reduced incorporation of the implant into surrounding tissue may thus be beneficial for the overall treatment with intramedullary nailing. Reduced ingrowth of material potentially results in condensed removal time, which would be favorable for both the patient and the health system in general.

Our model is focused on cell adhesion and differentiation on surface-modified titanium and steel plates. Further studies could be conducted to determine the influence of even deeper structures, wettability, chemical surface potentials, and the possible oxidization due to the laser process. Likewise, since the bone microenvironment is more complex than mesenchymal stem cells used in this study, further cell types such as osteoclasts, fibroblasts, or immune cells can be tested regarding their response to modified surfaces. Since the structure depth of periodic structures is about 1 µm and therefore close to the surface, influences on mechanical properties are not expected. Scratches of mechanical implants are around 10–20 µm, and even sandblasting creates deeper surface defects (~30 µm) [55].

For clinical use of modified implants, animal models need to be developed, where implant ingrowth, primary and secondary stability, fracture healing, and the immune response to modified titanium and steel implants can be analyzed.

## 5. Conclusions

In summary, our results demonstrate a modulation of human mesenchymal stem cells by controlling the surface characteristics via repetitive laser-induced nano-topographies. Reduced adhesion, altered cell morphologies, and reduced osteogenic differentiation capacities indicate that surface modification could be a suitable technique to locally reduce osseointegration in osteosynthetic materials such as titanium or steel implants. This may result in reduced removal times and material incorporation measurements, which would be beneficial for the patient and lower the cost for the healthcare system.

## Figures and Tables

**Figure 1 materials-13-03526-f001:**
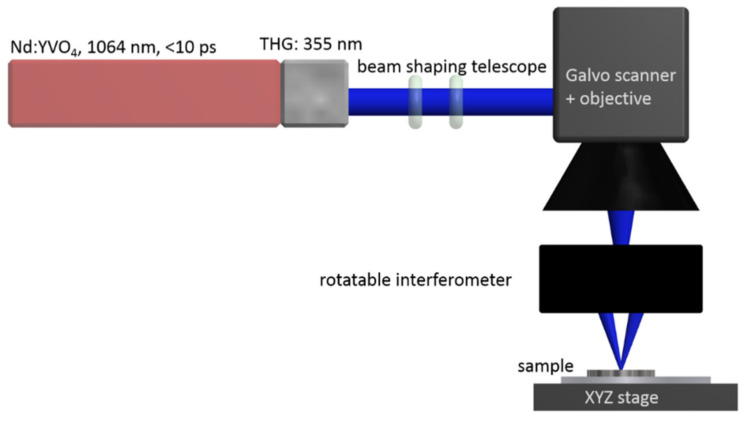
Setup for diffractive laser structuring of the implants. The galvo scanner moves the beam across the sample. The rotatable interferometer produces the basic line structures which can be overlaid with arbitrary angles by repeated radiation in order to produce more complex periodic structures.

**Figure 2 materials-13-03526-f002:**
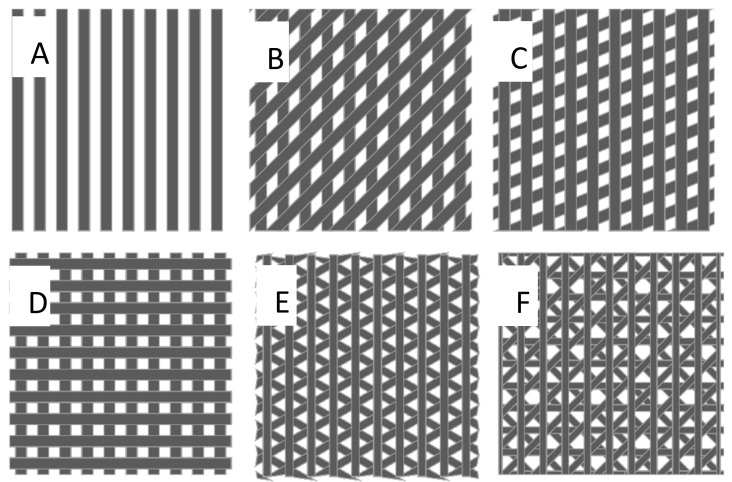
The six structure types. Linear (**A**), onefold crossed at angles of 45° (**B**), 67.5° (**C**), and 90° (**D**), twofold crossed (**E**) with crossing angles of 60° and 120°, threefold crossed (**F**) at 45°, 90° and 135°.

**Figure 3 materials-13-03526-f003:**
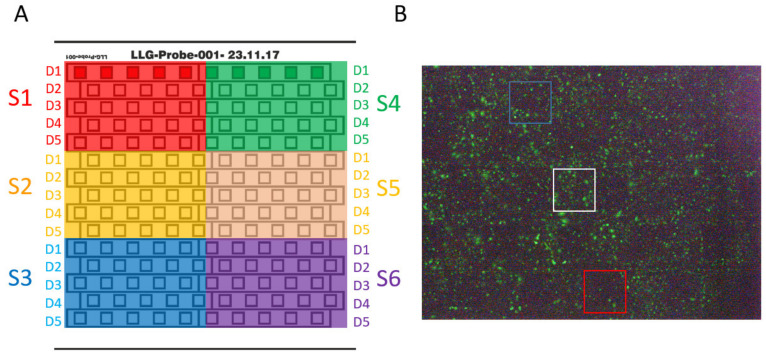
Sample preparation and examination. (**A**) Sample overview for the stainless-steel plates. The colored numbers outside of the square signify the six different structure types (S1–S6). Each structure is comprised of five columns and five rows, where the columns represent five different depths (D1–D5) and the rows are replicas for each structure/depth combinations. (**B**) Example of fluorescence microscopy image of structured (S5 D1–5) stainless steel plate. Fields with no structure (white), shallow (D1, blue), and deep structures (D5, red) are highlighted.

**Figure 4 materials-13-03526-f004:**
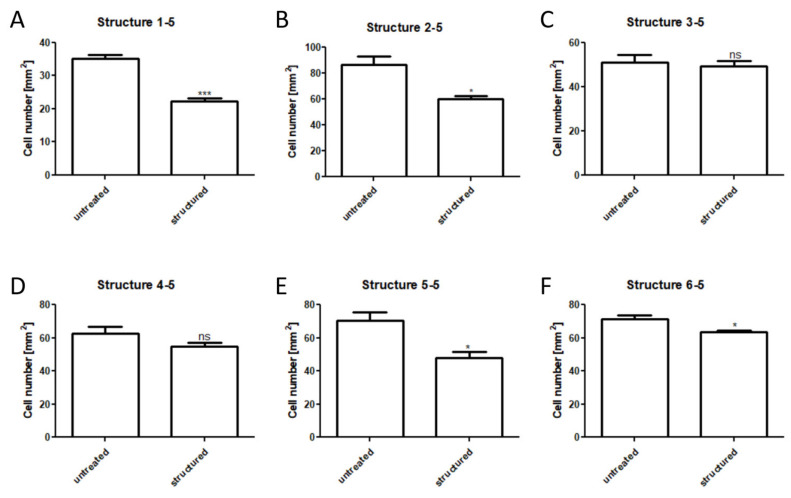
Analyzing cell adhesion on steel plates. Cell amount per mm^2^ was analyzed by fluorescence microscopy and manual cell counting after 48 h of cultivation. Significant reduction of cell adhesion was detected for the structures 1-5, 2-5, 5-5, and 6-5, while the structures 3-5 and 4-5 showed no significant differences. * (*p* < 0.05), *** (*p* < 0.0005), ns (non-significant).

**Figure 5 materials-13-03526-f005:**
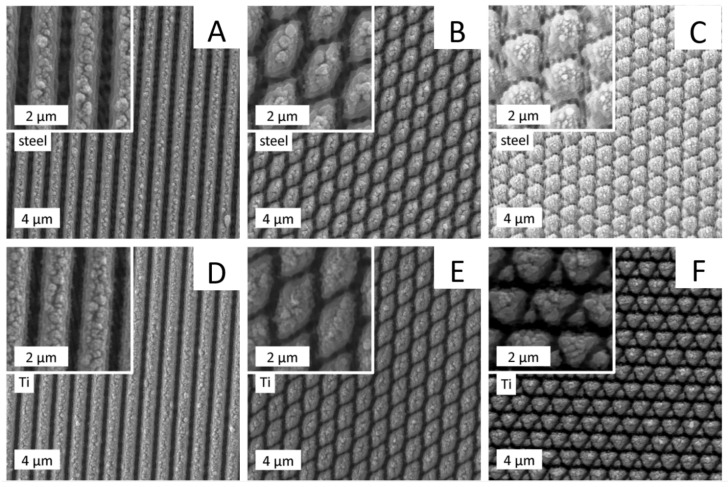
Scanning electron microscopy (SEM) examination of nanostructures. Stainless steel (**A**–**C**) and titanium (**D**–**F**) plates were analyzed using scanning electron microscopy. The following structures are shown: A and D: 1-5; B and E: 2-5; C and F: 5-5.

**Figure 6 materials-13-03526-f006:**
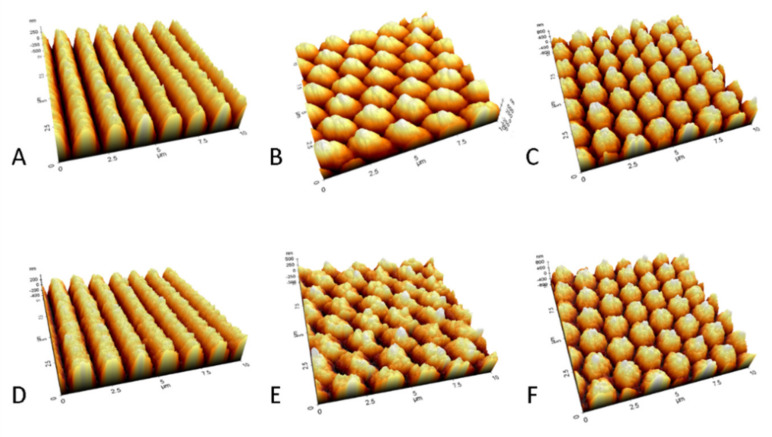
Surface topology analyzed by atomic force microscopy (AFM): (**A**–**C**): structures on stainless steel; (**D**–**E**): structures on titanium; A&D: structure 1-5; B&E: structure 2-5; C&F: structure 5-5. Stainless steel showed at least 100 nm deeper structures then titanium for these structure types.

**Figure 7 materials-13-03526-f007:**
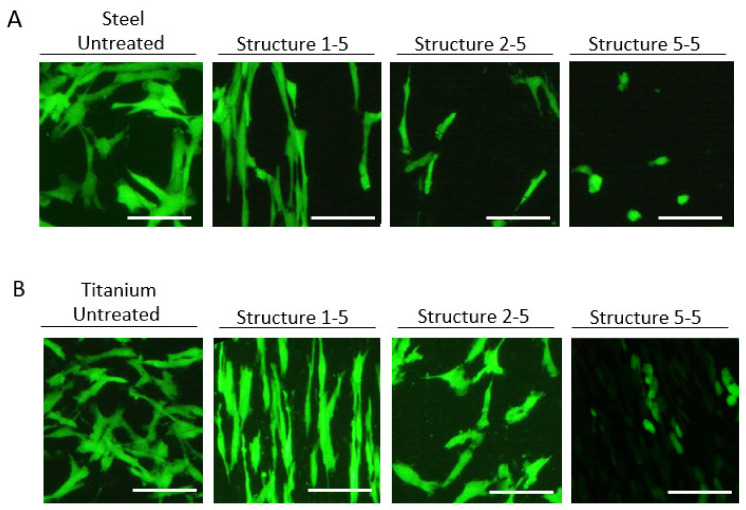
The influence of nanostructures on cell morphology. Whole plates of stainless steel (**A**) and titanium (**B**) were structured and SCP1 cell morphology was analyzed 48 h after cell seeding. The structure 1-5 lead to elongated and linearized cell morphology on titanium and steel plates. The structure 2-5 had only minor effects on cell morphology, while the structure 5-5 resulted in smaller and rounded cell phenotypes.

**Figure 8 materials-13-03526-f008:**
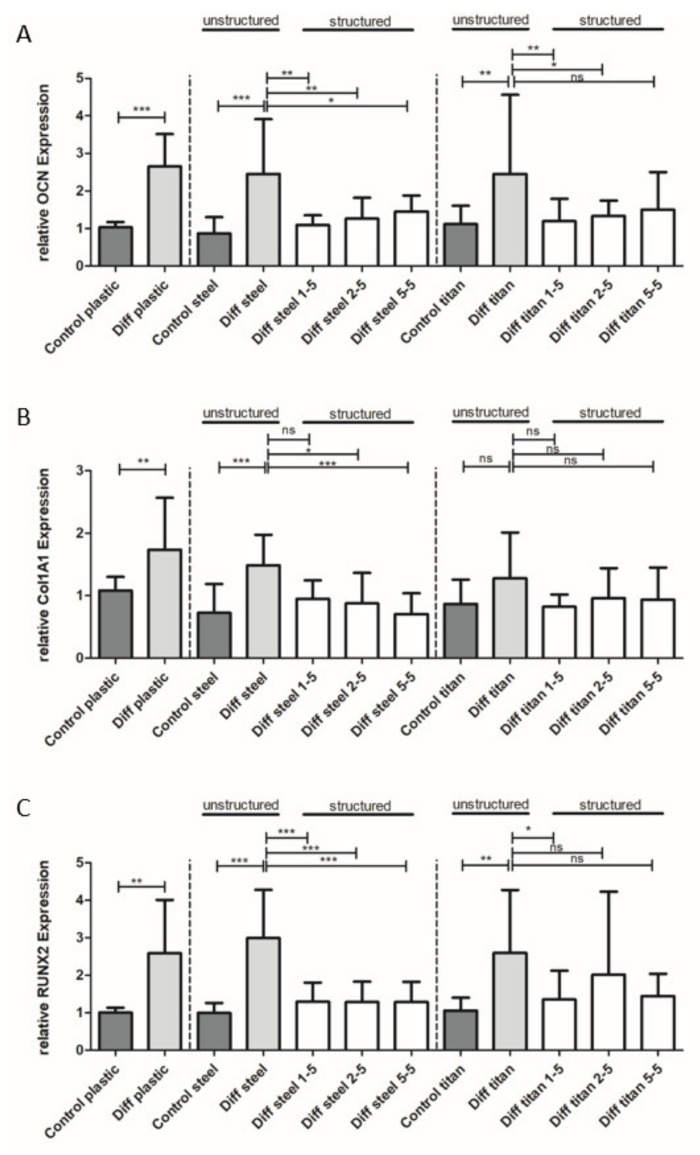
The influence of plate structuring on osteogenic differentiation. The gene expression analysis of osteogenic genes after 4 weeks of differentiation of human mesenchymal stem cells. Differentiation of MSCs according to gene expression was successful on polystyrene (plastic), steel, and titanium surfaces, while *Collagen 1* expression was not changed after differentiation on titanium plates. *OCN* (**A**) and *RUNX2* (**C**) expression was inhibited by all structured plates, while *Collagen 1* expression (**B**) was reduced on structured steel plates only. * (*p* < 0.05), ** (*p* < 0.005), *** (*p* < 0.0005), ns (non-significant).

**Table 1 materials-13-03526-t001:** Plasmids for lentiviral production (including depositor and addgene information).

Plasmid	Depositor	Addgene Number
psPAX2	Didier Trono	12,260
pLenti CMV GFP Neo	Eric Campeau [22]	17,447
pCMV-VSVG	Bob Weinberg [23]	8454

**Table 2 materials-13-03526-t002:** Osteogenic differentiation medium.

Component	Volume/Concentration	Company
DMEM (low glucose)	500 mL	Sigma Aldrich, Germany
Fetal calf serum (FCS)	50 mL (10%)	PAN Biotech, Germany
Penicillin/Streptavidin	5 mL (1%)	PAN Biotech, Germany
Ascorbic acid-2 phosphate	200 µM	Cayman chemical company, MI, USA
ß-glycerophosphate	10 mM	Carl Roth, Germany

**Table 3 materials-13-03526-t003:** Primer sequence.

Gene	Forward Primer (5´→3´)	Reverse Primer (5´→3´)
***β-2-microglobulin***	TGTGCTCGCGCTACTCTCTCT	CGGATGGATGAAACCCAGACA
***Osteocalcin***	GGCGCTACCTGTATCAATGG	GTGGTCAGCCAACTCGTCA
***Runx2***	TGGTTACTGTCATGGCGGGTA	TCTCAGATCGTTGAACCTTGCTA
***Collagen 1***	GTGCGATGACGTGATCTGTGA	CGGTGGTTTCTTGGTCGGT
***Collagen 2***	TGG ACG CCA TGA AGG TTT TCT	TGG GAG CCA GAT TGT CAT CTC
***Aggrecan***	GTGCCTATCAGGACAAGGTCT	GATGCCTTTCACCACGACTTC
***Sox9***	AGCGAACGCACATCAAGAC	CTGTAGGCGATCTGTTGGGG

**Table 4 materials-13-03526-t004:** Approximate depths of the deepest structures on steel and titanium measured by AFM. The structures 2-5 and 5-5 show multiple depths due to the multiple laser radiation in certain areas.

Material	Structure 1-5	Structure 2-5	Structure 5-5
**Stainless Steel**	880 nm	750 and 990 nm	1050, 1300 and 1400 nm
**Titanium**	700 nm	400 and 600 nm	900, 1200 and 1300 nm

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
