# Peer review of "Laser Ablated Periodic Nanostructures on Titanium and Steel Implants Influence Adhesion and Osteogenic Differentiation of Mesenchymal Stem Cells"

_materials, 2020, doi:10.3390/ma13163526_

Round 1

Reviewer 1 Report

'Laser ablated periodic nanostructures on titanium and steel implants influence adhesion and osteogenic differentiation of mesenchymal stem cells' review report:

This is an original study that aimed to reveal the impact of two material implants modified by laser on MSC cell line. It is adequately written, correctly sectioned and easy to understand. Experiment design is well conceived and in concordance with the study target. The results could have a significant impact for clinical practice. The authors also provide supplementary material which helps in the understanding of the paper. However, I found that some points that need discussion or modification. Therefore, I encourage authors to a minor revision of the following points;
- Please try to explain the abbreviations the first time they appear in the text (e.g. SEM or AFM)
- According to the structures created on the surfaces, what were the 5 different depths (D1-D5)?
- Some assays (Cell adhesion and morphology) the authors refer as a control 'unaltered' surface; however, in RNA expression analysis authors used as a comparator a 'standardized' surface. Could the authors clarify what the surface was as a comparison group?
- Cell adhesion and cell morphology are not clearly explained in the methodology section.
- Is there any reason why surfaces 3-5 and 4-5 were not morphology analyzed?
- The epigraph 'surface modifications inhibit osteogenic differentuatuion of human MSCs' should be replaced by 'surface modifications of human MSCs gene expression related to osteogenic inhibition'
- Please include all the genes analyzed in Figure 7
- Please thank Prof. Matthias Schieker for his collaboration in the ‘Acknowledgements’ section instead of in the methodology
- the SCP1 cell line and supplier needed to be described

- The methodology does not specify how the study of the cellular phenotype was carried out. I consider that the procedure that leads to the morphological study of the cells on each of the surfaces should be explained so that future work can replicate this test.

Reviewer 2 Report

The paper titled “Laser ablated periodic nanostructures on titanium and steel implants influence adhesion and osteogenic differentiation of mesenchymal stem cells” presents the effects of nanostructures created by laser beam interference ablation on mesenchymal stem cell adhesion and differentiation on metallic surfaces used for implants. The results show that some particular types of structures prevent cell adhesion and osteogenesis, showing that those structures could be used for implants that need to be eventually removed (nails, etc). The paper is well written and interesting.

My main concern lies with the number of cell experiments: how many independent experiments were performed? Technical replicates on their own are not convincing.

Regarding the discussion, are there no other reports on surface treatment methods to prevent osteogenesis? Many papers talk about methods to inhibit bacterial adhesion; has nothing been done with regards to mammalian cells? If so, this should be added in the discussion.

Also, gene names vary too much (for instance, collagen 1 is sometimes “collagen 1”, “Collagen1A1” or “Col1A1”) and should be standardized. Furthermore, official human gene names are usually in capital letters and italicized.

Some minor questions and suggestions are listed below:

L63: “has decreases” > “has decreased”

L84: “250 million Euro”: is that the figure for Germany, Europe, the “Western world” or worldwide?

L106: “30 combinations structures” > maybe “30 structure combinations” is better

L132-139: I suppose “untreated” is the area between the squares with no micro-structure; if so, why are the differences between the different “untreated” areas so large (for instance, 35 cells/mm2 for S1 and 86.6 cells/mm2 for S2)? How many times were the experiments performed independently? The 5 “replicas” (mentioned l119 and l127) are only technical replicates, and it is not clear what was used for the statistical analysis.

L151: “surface are of” > “surface of”

L169: “multiple laser radiation in certain areas”: do you mean that the radiation depends on the orientation of the line? Or that intersections are irradiated multiple times leading to deeper structures?

L248: “To test, if” > “To test if”

L275: “as well on” > “as well as on”

L284: “stronger effect” > “stronger effects”

L342: “methodical” and “clinical” > “methodically” and “clinically”

L369: “would beneficial” > “would be beneficial”

L433: the definition of SCP1 cells should come here, and not 3 paragraphs later, l459.

L445-446: “media” > “medium”

Author Response

The paper titled “Laser ablated periodic nanostructures on titanium and steel implants influence adhesion and osteogenic differentiation of mesenchymal stem cells” presents the effects of nanostructures created by laser beam interference ablation on mesenchymal stem cell adhesion and differentiation on metallic surfaces used for implants. The results show that some particular types of structures prevent cell adhesion and osteogenesis, showing that those structures could be used for implants that need to be eventually removed (nails, etc). The paper is well written and interesting.

Resonse: The authors would like to thank the referee for the very positive evaluation of our manuscript. We want to answer the comments of each section below:

My main concern lies with the number of cell experiments: how many independent experiments were performed? Technical replicates on their own are not convincing.

Response: Thank you for this important observation. We clarified the experimental procedure in the material and methods section (line 185). Cell adhesion experiments were performed in biological triplicates with 5 technical replicas each. For gene expression analysis, three technical replicates for each of the biological triplicate were performed (line 197, line 210).

Regarding the discussion, are there no other reports on surface treatment methods to prevent osteogenesis? Many papers talk about methods to inhibit bacterial adhesion; has nothing been done with regards to mammalian cells? If so, this should be added in the discussion.

Response:  We want to thank the reviewer for this important note. We added a paragraph about preventing osteogenesis in the discussion (line 431).

Also, gene names vary too much (for instance, collagen 1 is sometimes “collagen 1”, “Collagen1A1” or “Col1A1”) and should be standardized. Furthermore, official human gene names are usually in capital letters and italicized.

Response: We have now standardized the gene names of Collagen 1 in the whole manuscript and added italicized letters.

Some minor questions and suggestions are listed below:

L63: “has decreases” > “has decreased”

Response: We agree with the reviewer´s comment and corrected the sentence.

L84: “250 million Euro”: is that the figure for Germany, Europe, the “Western world” or worldwide?

Response: We thank the reviewer for pointing out that this is not yet clearly described. We clarified the sentence and described that these numbers are summarizing the costs for Germany (line 83).

L106: “30 combinations structures” > maybe “30 structure combinations” is better

Response: We want to thank the reviewer for this suggestion. We have modified the sentence accordingly.

L132-139: I suppose “untreated” is the area between the squares with no micro-structure; if so, why are the differences between the different “untreated” areas so large (for instance, 35 cells/mm2 for S1 and 86.6 cells/mm2 for S2)? How many times were the experiments performed independently? The 5 “replicas” (mentioned l119 and l127) are only technical replicates, and it is not clear what was used for the statistical analysis.

Response: We want to thank the reviewer for this important question. All cell culture experiments were performed with 3 biological replicas each. For cell adhesion, 5 technical replicas were used, while gene expression analysis was performed with 3 technical replicas each. This fact was clarified in the materials and method section. Regarding the different cell densities, we observed different cell densities on the structured plates (Figure 3B, lower cell density on the right/down right side). That’s why we choose the untreated area directly beside the structured area, to limit variations during cell counting. We have now clarified the procedure in 2.6 Cell adhesion determination (line 185).

L151: “surface are of” > “surface of”

Response: We agree with this reviewer point. We changed the sentence to clarify that the whole plate was modified by laser treatment.

L169: “multiple laser radiation in certain areas”: do you mean that the radiation depends on the orientation of the line? Or that intersections are irradiated multiple times leading to deeper structures?

Response: We thank the reviewer for this comment. We meant the intersections of the overlaying line structures where the structural depth is deeper than in areas where the lines do not intersect. We modified the sentence accordingly (line 280).

L248: “To test, if” > “To test if”

Response: We corrected the sentence.

L275: “as well on” > “as well as on”

Response: We agree with the reviewer’s point and modified the sentence.

L284: “stronger effect” > “stronger effects”

Response: We modified as proposed by the reviewer.

L342: “methodical” and “clinical” > “methodically” and “clinically”

Response: We adjusted the sentence as mentioned.

L369: “would beneficial” > “would be beneficial”

Response: We corrected the sentence accordingly.

L433: the definition of SCP1 cells should come here, and not 3 paragraphs later, l459.

Response: We moved the definition of SCP1 cells ad proposed and modified the text according to the reviewer’s suggestion.

L445-446: “media” > “medium”

Response: We changed the paragraph according to the reviewer’s suggestion.

Reviewer 3 Report

First of all, please prepare your article according to the template and instructions for authors: font, punctuation, line spacing, indents, subsections numbering etc.

Please have the manuscript further checked for English language, there are a number of expression mistakes.

Please give abbreviations in full when used first time. Do not give them separately in the end.

Do not give long explanations as figure captions.

Introduction:

Line 56 "Currently, alloys for such intramedullary nails are made of titanium or stainless steel" Please rephrase: "implants for such intramedullary nails" would be better.

Material and methods:

Please make it chapter 2, not 4.

The only information regarding the samples is given in line 419: "For the initial testing, the polished stainless steel samples (14x14mm) were provided with 0.5×0.5 mm square test fields." The samples should be described in the beginning of the M&M chapter. What about the titanium ones, no information is available. What do you mean by polished? In what degree of roughness (mirror polished or otherwise...)

Some images of the samples should be provided to document your work.

Table 2. In my opinion, the name of the persons who gifted you materials do not belong in the paper, there is a special acknowledgements section in the end.

Results:

The subsection Surface topography, line 104-114 better fits to the M&M chapter. Pictures of the structures would be appropriate, instead of  schemes.

Line 150: "...and 14×14 mm polished stainless steel as well as titanium samples were structured on the full surface are of 14 x 14 mm" This is not understandable.

Conclusion:

A conclusion section would be appropriate.

Author Response

First of all, please prepare your article according to the template and instructions for authors: font, punctuation, line spacing, indents, subsections numbering etc.

Response: We want to thank the reviewer for this important note. The whole article was organized according to the MDPI template file (line spacing, font, indents).

Please have the manuscript further checked for English language, there are a number of expression mistakes.

Response: We want to thank the reviewer for this observation. We checked the whole manuscript for English language and highlighted the changes in yellow.

Please give abbreviations in full when used first time. Do not give them separately in the end.

Response: Thank you for this suggestion. We give all abbreviations in full when used the first time in the main document and removed the abbreviations at the end.

Do not give long explanations as figure captions.

Response: We want to thank the reviewer for this comment. The explanations for the figure captions were shortened for several figures (e.g. figure 7 and figure 8)

Introduction:

Line 56 "Currently, alloys for such intramedullary nails are made of titanium or stainless steel" Please rephrase: "implants for such intramedullary nails" would be better.

Response: We adjusted the mentioned sentence (line 56).

Material and methods:

Please make it chapter 2, not 4.

Response: We want to thank the reviewer for this comment. The whole article was adapted to the MDPI guidelines and the material and method section was moved to chapter 2.

The only information regarding the samples is given in line 419: "For the initial testing, the polished stainless steel samples (14x14mm) were provided with 0.5×0.5 mm square test fields." The samples should be described in the beginning of the M&M chapter. What about the titanium ones, no information is available. What do you mean by polished? In what degree of roughness (mirror polished or otherwise...)

Response: We agree with the reviewers comment. Material and method section was moved to chapter 2. Surface structuring of all samples is now described in the beginning of M&M (line 103). The description of complete structuring of titanium and steel plates was added (line 143) and polishing is now described (line 146). We added a supplementary figure for a better understanding (see next comment)

Some images of the samples should be provided to document your work.

Response: We added a supplementary figure (Supplementary Figure S1) with images of structured plates to document our work as proposed by the reviewer.

Table 2. In my opinion, the name of the persons who gifted you materials do not belong in the paper, there is a special acknowledgements section in the end.

Response: Thank you for this suggestion. We renamed parts of table 1 according to the reviewers suggestion. According to the addgene guidelines, depositor and addgene number have to stay in material and methods (line 177). We hope this compromise is accepted.

Results:

The subsection Surface topography, line 104-114 better fits to the M&M chapter. Pictures of the structures would be appropriate, instead of schemes.

Response: Thank you for this comment. We agree with your assessment and moved the content of this subsection to the M&M part “2.1 Surface structuring of implant materials” (line 137).

Line 150: "...and 14×14 mm polished stainless steel as well as titanium samples were structured on the full surface are of 14 x 14 mm" This is not understandable.

Response: Thank you for detecting this small inaccuracy. We clarified the structuring of the steel plates with 30 structures (Figure 3) and the complete structuring of steel and titanium plates in the text and with an additional supplementary figure (line 123, line 143 and Supplementary Figure S1).

Conclusion:

A conclusion section would be appropriate.

Response: We added a conclusion section at the end of the discussion (line 460).

Round 2

Reviewer 3 Report

Figure 5: Scannin electron microscopy. Please correct.

Thank you for your answers. I have no further comments.

Author Response

We want to thank the reviewer for the time dedicated to the evaluation of our manuscript. The critics were very useful to improve the manuscript.

We corrected the sentence in the manuscript (line 261).